# Associations of antithrombotic agent use with clinical outcomes in critically ill patients with troponin I elevation in the absence of acute coronary syndrome

Chuan-Tsai Tsai[1,2☯], Ya-Wen Lu[1,2☯], Ruey-Hsing Chou[1,2,3,4], Yi-Lin Tsai[1,2], Ming-Ren Kuo[1,2], Jiun-Yu Guo[2], Chi-Ting Lu[2], Chin-Sung Kuo[2,4,5], Po-Hsun Huang[1,2,3,4]*

1 Division of Cardiology, Department of Medicine, Taipei Veterans General Hospital, Taipei, Taiwan, 2 Cardiovascular Research Center, National Yang-Ming University, Taipei, Taiwan, 3 Department of Critical Care Medicine, Taipei Veterans General Hospital, Taipei, Taiwan, 4 Institute of Clinical Medicine, National Yang-Ming University, Taipei, Taiwan, 5 Division of Endocrinology and Metabolism, Department of Medicine, Taipei Veterans General Hospital, Taipei, Taiwan

☯ These authors contributed equally to this work.
* huangbsvgh@gmail.com

**Data Availability Statement:** All data files are available from the figshare database (accession number 10.6084/m9.figshare.12154896).

## Abstract

### Introduction

To evaluate efficacy of antithrombotic agents in critically ill patients with elevated troponin I level during intensive care unit (ICU) admission.

### Methods and results

It was a retrospective observational study which was conducted in a tertiary teaching hospital in Taipei, Taiwan. All patients hospitalized in ICU for >3 days and with available serum troponin I data from December 2015 to July 2017 were included. Patients with definite diagnosis of acute myocardial infarction (AMI) were excluded. We divided patients with troponin I elevation into three groups; no prescription, chronic prescription and new prescription of antithrombotic agents during ICU admission. We defined new prescription when patients were on antithrombotic agents, including antiplatelet agents, direct oral anticoagulants, and warfarin after troponin I was found to be elevated at ICU admission and chronic prescription, if antithrombotic agents were on medication list more than 30 days before ICU admission. Primary outcomes were 30-day and one-year all-cause mortality. Of 597 subjects who met inclusion criteria, 407 (68%) patients had elevated troponin I (>0.1 ng/mL) on ICU admission. These patients had increased 30-day [hazard ratio (HR), 1.679; 95% confidence interval (CI), 1.132–2.491; $p = 0.009$] and one-year (HR, 1.568; 95% CI, 1.180–2.083; $p = 0.002$) all-cause mortality compared with those without elevated troponin I. In patients with elevated troponin I, there was no significant difference of 30-day all-cause mortality among three groups ($p = 0.051$) whereas patients on chronic prescription showed significant survival benefit in one-year all-cause mortality when compared to those without or with new prescription ($p = 0.008$).

**Funding:** This study was supported, in part, by research grants from the Ministry of Science and Technology of Taiwan (MOST 106-2314-B-350-001-MY3); the Novel Bioengineering and Technological Approaches to Solve Two Major Health Problems in Taiwan program, sponsored by the Taiwan Ministry of Science and Technology Academic Excellence Program (MOST 108-2633-B-009-001); the Ministry of Health and Welfare (MOHW106-TDU-B-211-113001); and Taipei Veterans General Hospital (V105C-207, V106C-045, V108C-195). These funding agencies had no influence on the study design, data collection or analysis, decision to publish, or preparation of the manuscript.

**Competing interests:** No authors have competing interests.

## Conclusions

In critically ill patients, elevated troponin I in the absence of AMI was associated with poor prognosis. Newly prescribed antithrombotic agents in ICU didn't reveal the difference in short and long-term prognosis while chronic antithrombotic agent use was associated with better one-year survival rate, suggesting that these drugs play a protective role in this high-risk population.

## Introduction

Acute myocardial infarction (MI) is diagnosed with a combination of clinical symptoms, signs, electrocardiography, and acute elevation of biomarkers, such as creatinine kinase and troponin I. [1] Troponin I that was elevated due to myocardial injury from thrombus occlusion of epicardial coronary artery was defined as type 1 MI. However, troponin I was frequently found to be elevated from the conditions known as type 2 MI. It was related to myocardial oxygen supply and demand mismatch, coronary hypoperfusion from generalized hypotension such as septic shock, respiratory failure, pulmonary embolism and acute heart failure. [2] Previous studies found that patients with type 2 MI when admitted to intensive care unit (ICU) was associated with higher in-hospital mortality compared to those with type 1 MI. [3–5] Due to differences in the mechanisms that lead to myocardial injury, the definitive treatment for type 2 MI is an essential clinical issue. [5]

Aspirin reduced all-cause mortality by about one-sixth and vascular death by 15% in patients with occlusive coronary artery disease (i.e., type 1 MI) and multiple risk factors for atherosclerosis, such as diabetes mellitus, ischemic stroke, and peripheral arterial disease. [6] Adverse outcomes were further reduced by combination therapy with aspirin and P2Y12 inhibitors, such as clopidogrel and ticagrelor, in patients with type 1 MI. [7] However, the exact role of antithrombotic agents in patients with type 2 MI remains uncertain. We thus conducted this retrospective study to investigate the impact of these medications in patients with type 2 MI when admitted to non-cardiac ICU.

## Methods

### Study population

For this retrospective study, we screened 2678 patients admitted to the ICU at Taipei Veterans General Hospital due to critical conditions and the requirement for intensive care between December 2015 and July 2017. Patients were admitted from the emergency room or transferred from the ordinary ward to the medical or surgical ICU (SICU). Patients with available serum troponin I at ICU admission were enrolled. We excluded patients whose ICU stays were <3 days, those undergoing postsurgical monitoring or post–percutaneous coronary intervention monitoring and those with definite diagnosis of type 1 MI, i.e., symptoms of angina, plus new ischemic ECG changes and identification of thrombus occlusion of coronary artery by angiography. This study was conducted according to the principles of the Declaration of Helsinki and was approved by the Research Ethics Committee of Taipei Veterans General Hospital. All participants provided written informed consent. A flowchart of patient enrollment and classification is shown in Fig 1.

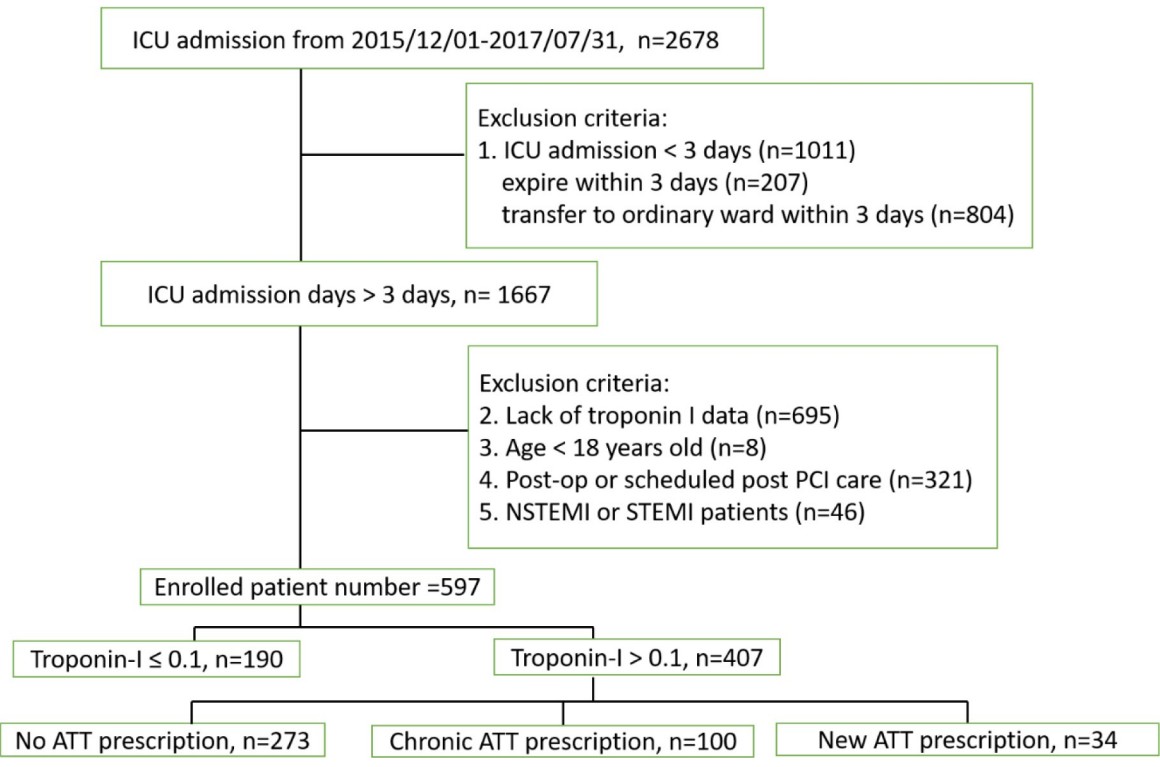

**Fig 1. Flow chart of patients enrollment.** dx.doi.org/10.17504/protocols.io.bfcajise.

## Measurement of clinical variables

We reviewed the medical records of each patient in detail after enrollment. Data were collected on patients' clinical characteristics, including age, sex, body weight, body mass index (BMI), comorbidities, etiologies of ICU admission, categories of chronic medication use, and disease severity. The acute physiology and chronic health evaluation II (APACHE II) and Sequential Organ Failure Assessment (SOFA) scores were calculated to evaluate disease severity in the first 24 hours after ICU admission. [8, 9] We reviewed clinical and microbiological data to collect information on infection foci. Blood chemistry parameters were measured and cell counts were determined by routine laboratory methods on the first day of ICU admission. The estimated glomerular filtration rate was calculated using the Modification of Diet in Renal Disease equation. Serum troponin I level was measured by electrochemiluminescence immunoassay (Elecsys; Roche Diagnostics GmbH, Mannheim, Germany) using Cobas e 600 analyzer, for which the 99[th] percentile of the upper reference limit value is 0.10 ng/mL. We stratified the study cohort according to the troponin I value, and compared the baseline characteristics of study subjects with troponin I levels > 0.10 ng/mL with those with lower serum troponin I levels.

## Identification of exposure to antithrombotic agents

We retrieved data on antithrombotic use from patients' electronic medical records. We divided patients with elevated troponin I into three groups according to anticoagulant medication; no, chronic and new prescription. We defined the chronic use of antithrombotic agents, including antiplatelet agents (e.g., aspirin, clopidogrel, ticlopidine, dipyridamole), direct oral anticoagulants, and vitamin K antagonists (e.g., warfarin) as prescription >30 days before ICU

admission. We also recorded new prescriptions and withdrawals of antithrombotics during the patients' ICU stays.

## Study outcomes and patient follow-up

We recorded short and long-term adverse events for the study sample. Short-term adverse events included the lengths of ICU stay and hospitalization, gastrointestinal tract bleeding during the ICU stay, acute renal failure requiring dialysis during the ICU stay, and in-hospital and 30-day mortality. Long-term adverse events included the prevalence of persistent ventilator dependence, dialysis dependence, and one-year mortality. The primary outcomes were 30-day and one-year mortality. Respiratory failure was defined as the need for endotracheal intubation and mechanical ventilator support. [10] Acute kidney injury (AKI) was defined as the acute deterioration of renal function and requirement for renal replacement therapy, according to the Acute Kidney Injury Network criteria. [11] Sepsis was defined as life-threatening organ dysfunction caused by dysregulated host response to infection, as reflected by a ≥2-point increase in the SOFA score. [12] Patients were followed until December 2018, with all-cause mortality, rehospitalization, and persistent ventilator or dialysis dependence recorded and analyzed.

## Statistical analysis

Categorical variables were expressed as numbers and percentages and compared using the chi-squared test. Continuous variables were expressed as means ± standard deviations and compared between groups using analysis of variance or the Mann–Whitney $U$ test. For the comparison of survival rates of patients with different troponin I levels and the effect of antithrombotics among those with higher troponin levels, we used Kaplan–Meier survival curves and the log-rank test. Cox proportional-hazard regression analyses were performed to obtain hazard ratios (HRs) and associated 95% confidence intervals (CIs). Variables with $p$ values $< 0.05$ in the univariable regression analysis were entered in the multivariable regression analysis. Multivariable Cox regression analyses were performed for the whole study cohort and for patients with higher troponin I levels. Interactions between variables and chronic antithrombotic use were examined for one-year mortality. $P$ values $< 0.05$ were considered to be significant. All analyses were performed using SPSS software (version 19.0; IBM Corporation, Armonk, NY, USA).

# Results

## Baseline characteristics and clinical outcomes stratified by troponin I level

In total, 597 study subjects admitted to the ICU for >3 days with available serum troponin I data were included in this study; 407 (68%) of these patients had elevated troponin I (> 0.1 ng/ mL) at ICU admission. Baseline characteristics of the study cohort are provided in Table 1. Patients in the elevated troponin I group had higher APACHE II and SOFA scores, more hemodynamic instability, less percentage of post-operation, higher serum creatinine and lactate level, and lower platelet counts than did the rest of the cohort. Among underlying diseases, end-stage renal disease and chronic heart failure were more prevalent among patients in the elevated troponin I group.

## Troponin I elevation was associated with increased 30-day and one-year mortality

Short and long-term outcomes are summarized in Table 2. A total of 209 (35%) patients died during hospitalization. Compared with critically ill patients without elevated troponin I levels,

**Table 1. General features of the patients, according to serum troponin I levels at ICU admission.** dx.doi.org/10.17504/protocols.io.bfbejije.

| | Overall (n = 597) | Troponin I ≤ 0.1 (n = 190) | Troponin I > 0.1 (n = 407) | *P* value |
|---|---|---|---|---|
| Age (year) | 70 ± 17 | 70 ± 16 | 70 ± 17 | 0.861 |
| Sex (male) | 385 (65) | 129 (68) | 256 (63) | 0.271 |
| APACHE II | 27 ± 8 | 26 ± 8 | 28 ± 8 | <0.001 |
| SOFA | 9 ± 3 | 8 ± 3 | 10 ± 3 | <0.001 |
| Admission diagnosis | | | | |
| Respiratory failure (%) | 311 (52) | 104 (55) | 207 (51) | 0.381 |
| hemodynamic unstable (%) | 176 (30) | 40 (21) | 136 (33) | 0.002 |
| Post operation (%) | 64 (11) | 35 (18) | 29 (7) | <0.001 |
| Focus of infection | | | | |
| Pneumonia (%) | 290 (49) | 95 (50) | 195 (48) | 0.661 |
| Catheter related infection (%) | 65 (11) | 17 (9) | 48 (12) | 0.326 |
| Intraabdominal infection (%) | 90 (15) | 35 (18) | 55 (14) | 0.140 |
| Lab findings, Mean ± SD | | | | |
| Troponin I (ng/ml) | 1.32 ± 3.90 | 0.1 ± 0.0 | 1.9 ± 4.6 | <0.001 |
| Serum creatinine (mg/dl) | 2.69 ± 2.20 | 2.34 ± 2.09 | 2.86 ± 2.23 | 0.009 |
| HCO3 (mmHg) | 20.9 ± 5.0 | 21.7 ± 4.7 | 20.6 ± 5.0 | 0.013 |
| Lactate (mg/dl) | 25 ± 26 | 17 ± 15 | 29 ± 29 | <0.001 |
| CRP (mg/dl) | 11.9 ± 5.4 | 14.5 ± 11.3 | 12.3 ± 10.7 | 0.172 |
| Underlying disease | | | | |
| Hypertension (%) | 319 (53) | 105 (55) | 214 (53) | 0.597 |
| Diabetes mellitus (%) | 221 (37) | 67 (35) | 154 (38) | 0.585 |
| COPD (%) | 30 (5) | 10 (5) | 20 (5) | 0.843 |
| ESRD (%) | 62 (10) | 10 (5) | 52 (13) | 0.006 |
| CHF (%) | 80 (13) | 17 (9) | 63 (16) | 0.029 |
| Prior history of CAD (%) | 116 (19) | 37 (20) | 79 (20) | 1.000 |
| Prior stroke (%) | 48 (8) | 15 (8) | 33 (8) | 1.000 |
| Cancer (%) | 151 (25.3) | 61 (32.1) | 90 (22.1) | 0.001 |
| Chronic medication use | | | | |
| ACEI/ARB | 142 (23.8) | 53 (27.9) | 89 (21.9) | 0.122 |
| beta-blocker | 117 (19.6) | 40 (21.1) | 77 (18.9) | 0.580 |
| Statin | 76 (12.7) | 20 (10.5) | 56 (13.8) | 0.294 |
| OHA | 115 (19.3) | 26 (13.7) | 89 (21.9) | 0.019 |
| ATT | 148 (24.8) | 48 (25.3) | 100 (24.6) | 0.539 |
| ATT during ICU admission | 224 (37.5) | 66 (34.7) | 158 (38.8) | 0.365 |

Values are given as mean and standard deviation or numbers and percentages.

APACHE, acute physiology and chronic health evaluation; SOFA, sequential organ failure assessment; CRP, C reactive protein; COPD, chronic obstructive pulmonary disease; ESRD, end stage renal disease; CHF, congestive heart failure; CAD, coronary artery disease; ACEI, angiotensin converting enzyme inhibitor; ARB, angiotensin receptor blocker; OHA, oral hypoglycemic agents; ATT, anti-thrombotics.

those with troponin I levels > 0.1 ng/mL had higher incidences of in-hospital mortality (27% vs. 39%, *p* = 0.008), 30-day mortality (17% vs. 27%, *p* = 0.007), and one-year mortality (34% vs. 46%, *p* = 0.006). The incidence of long-term ventilator and dialysis dependence did not differ between groups.

The survival rates of patients, stratified by troponin I level, are shown in Fig 2. The 30-day and one-year all-cause mortality rates were significantly higher in patients with elevated troponin I level. In the multivariate Cox regression analysis of all patients admitted to ICU adjusted

**Table 2. Various outcomes according to troponin I level.** dx.doi.org/10.17504/protocols.io.bfbhjij6.

| | Overall (n = 597) | Troponin I ≤ 0.1 (n = 190) | Troponin I > 0.1 (n = 407) | *P* value |
|---|---|---|---|---|
| **Short-term outcomes** | | | | |
| Length of ICU stay | 11 ± 7.0 | 11 ± 7.2 | 11 ± 7.0 | 0.939 |
| Length of hospitalization | 36 ± 37.2 | 38 ± 32.5 | 36 ± 39.3 | 0.510 |
| GI bleeding during admission | 394 (66.1) | 123 (65.1) | 271 (66.6) | 0.780 |
| AKI during admission (%) | 243 (41) | 76 (40) | 167 (41) | 0.858 |
| Hospital mortality (%) | 209 (35) | 52 (27) | 157 (39) | 0.008 |
| 30-day mortality (%) | 142 (24) | 32 (17) | 110 (27) | 0.007 |
| **Long-term outcomes** | | | | |
| Ventilator dependent (%) | 33 (6) | 10 (5) | 23 (6) | 1.000 |
| Dialysis dependent (%) | 64 (11) | 16 (8) | 48 (12) | 0.256 |
| One-year mortality (%) | 251 (42) | 64 (34) | 187 (46) | 0.006 |

Values are given as mean and standard deviation or numbers and percentages.

GI, gastrointestinal; AKI, acute kidney injury

for the APACHE II score, diagnosis at admission, infection focus, serum lactate level, underlying disease with cancer, and chronic medication (beta-blockers and antithrombotics) use, troponin I > 0.1 ng/mL was associated independently with one-year mortality (Table 3).

## Antithrombotic use during ICU stays in patients with troponin I elevation

In 407 patients with elevated troponin I, 273 (67%) patients were not on antithrombotics while 100 (25%) patients were on chronic prescription and 34 (8%) patients on new prescription during ICU admission. Patients using chronic antithrombotics were older, with higher

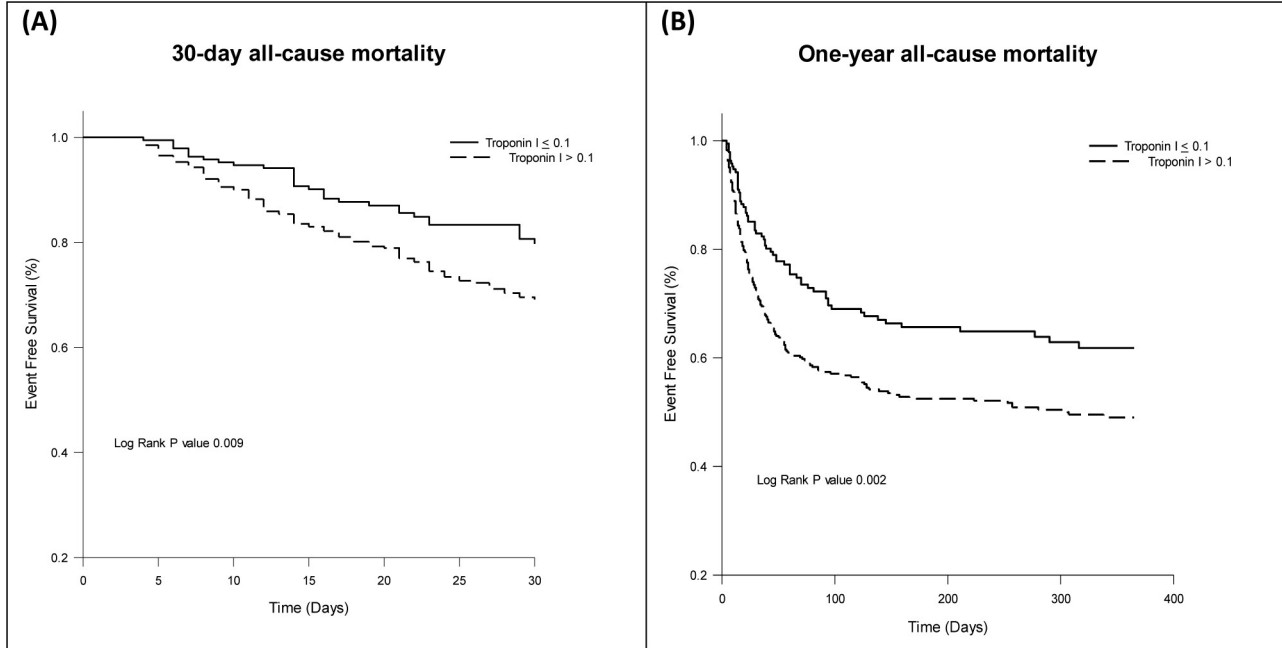

**Fig 2.** Kaplan-Meier curve for all-cause mortality according to troponin I level at ICU admission (A) 30-day all-cause mortality (B) one-year all-cause mortality. dx.doi.org/10.17504/protocols.io.bfbqjimw.

**Table 3. Multivariate analysis of risk of one-year mortality of all patients by a Cox regression model.** dx.doi.org/10.17504/protocols.io.bfbijike.

|  | Univariate analysis | | Multivariate analysis | |
|  | HR (95% CI) | P value | HR (95% CI) | P value |
| --- | --- | --- | --- | --- |
| Age (year) | 1.003 (0.995–1.010) | 0.469 | | |
| Sex (male) | 1.161 (0.893–1.509) | 0.264 | | |
| APACHE II | 1.064 (1.047–1.081) | <0.001 | 1.062 (1.042–1.083) | <0.001 |
| SOFA | 1.123 (1.081–1.166) | <0.001 | | |
| Admission diagnosis | | | | |
| Respiratory failure | - | - | - | - |
| hemodynamic unstable | 1.348 (1.036–1.755) | 0.026 | 1.225 (0.907–1.655) | 0.185 |
| Post operation | 0.367 (0.230–0.586) | <0.001 | 0.569 (0.340–0.952) | 0.032 |
| Focus of infection | | | | |
| Pneumonia | 1.604 (1.248–2.061) | <0.001 | 1.458 (1.080–1.968) | 0.014 |
| Bacteremia | 1.208 (0.829–1.759) | 0.326 | | |
| Intraabdominal infection | 0.910 (0.640–1.296) | 0.602 | | |
| Lab findings | | | | |
| Troponin I >0.1 (ng/ml) | 1.568 (1.180–2.083) | 0.002 | 1.516 (1.096–2.095) | 0.012 |
| Serum creatinine (mg/dl) | 0.967 (0.910–1.027) | 0.277 | | |
| HCO3 | 1.020 (0.995–1.046) | 0.118 | | |
| Lactate (mg/dl) | 1.005 (1.001–1.009) | 0.028 | 0.997 (0.992–1.003) | 0.357 |
| CRP (mg/dl) | 0.991 (0.973–1.010) | 0.347 | | |
| Underlying disease | | | | |
| Hypertension | 0.917 (0.716–1.175) | 0.493 | | |
| Diabetes mellitus | 0.898 (0.693–1.164) | 0.417 | | |
| ESRD | 0.957 (0.633–1.447) | 0.837 | | |
| CHF | 1.280 (0.916–1.789) | 0.148 | | |
| CAD | 0.907 (0.662–1.243) | 0.543 | | |
| CVD | 0.673 (0.406–1.117) | 0.126 | | |
| Cancer | 1.428 (1.092–1.868) | 0.009 | 1.684 (1.252–2.265) | 0.001 |
| ACEI/ARB, chronic | 0.961 (0.719–1.285) | 0.788 | | |
| Beta-blocker, chronic | 0.519 (0.360–0.749) | <0.001 | 0.600 (0.392–0.917) | 0.018 |
| Statin, chronic | 0.767 (0.514–1.143) | 0.193 | | |
| OHA, chronic | 1.018 (0.741–1.397) | 0.914 | | |
| ATT, chronic prescription | 0.574 (0.417–0.790) | 0.001 | 0.639 (0.446–0.916) | 0.015 |
| ATT, new prescription | 0.898 (0.582–1.385) | 0.626 | | |

HR, hazard ratio; CI, confidence interval; APACHE, acute physiology and chronic health evaluation; SOFA, sequential organ failure assessment; CRP, C reactive protein; ESRD, end stage renal disease; CHF, congestive heart failure; CAD, coronary artery disease; CVD, cerebrovascular disease; ACEI, angiotensin converting enzyme inhibitor; ARB, angiotensin receptor blocker; OHA, oral hypoglycemic agent; ATT, anti-thrombotics; ICU, intensive care unit.

prevalence of hypertension, diabetes mellitus, congestive heart failure, coronary artery disease (CAD) and concomitant medication with angiotensin converting enzyme inhibitor, angiotensin receptor blocker, beta blocker, statin and oral hypoglycemic agents. Patients received new prescription showed higher level of initial troponin I. No difference was observed in the percentages of ICU admission causes, infection focus, or serum parameters other than troponin I (including the blood urea nitrogen level; platelet count; pH; and bicarbonate, lactate, and C-reactive protein levels among three groups (Table 4). Subsequent coronary revascularization was performed highest in patients with chronic antithrombotic medication although statistically not significant. The lengths of ICU stay and hospitalization did not differ among three

**Table 4. General features of the patients, according to medications in patients with elevated troponin I.** dx.doi.org/10.17504/protocols.io.bfbjjikn.

| | No prescription (n = 273) | Chronic prescription (n = 100) | New prescription (n = 34) | *P* value |
|---|---|---|---|---|
| Age (year) | 67 ± 18 | 76 ± 12 | 69 ± 17 | 0.0001 |
| Sex (male) | 171 (63) | 65 (65) | 20 (59) | 0.803 |
| APACHE II | 28 + 8 | 28 + 8 | 28 + 7 | 0.853 |
| SOFA | 9.7 + 3.5 | 9.7 + 3.2 | 9.3 + 3.2 | 0.877 |
| Coronary revascularization (%) | 17 (6) | 13 (13) | 1 (3) | 0.052 |
| Admission diagnosis | | | | |
| Respiratory failure (%) | 144 (53) | 49 (49) | 14 (41) | 0.406 |
| hemodynamic unstable (%) | 91 (33) | 34 (34) | 11 (32) | 0.983 |
| Post op (%) | 38 (14) | 16 (16) | 9 (27) | 0.160 |
| Focus of infection | | | | |
| Pneumonia (%) | 132 (48) | 47 (47) | 16 (47) | 0.968 |
| Catheter related infection (%) | 34 (13) | 8 (8) | 6 (18) | 0.270 |
| Intraabdominal infection (%) | 40 (15) | 12 (12) | 3 (9) | 0.566 |
| Lab findings, Mean ± SD | | | | |
| Troponin I (ng/ml) | 1.36 + 2.81 | 2.16 + 4.13 | 5.40 + 11.4 | 0.0001 |
| Serum creatinine (mg/dl) | 2.8 + 2.2 | 3.2 + 2.4 | 2.2 + 1.2 | 0.091 |
| HCO3 (mmHg) | 20.5 + 5.3 | 20.5 + 4.6 | 21.2 + 4.2 | 0.756 |
| Lactate (mg/dl) | 31.1 + 31.7 | 23.8 + 20.2 | 22.4 + 24.0 | 0.050 |
| CRP (mg/dl) | 11 + 10 | 16 + 12 | 12 + 12 | 0.106 |
| Underlying disease | | | | |
| Hypertension (%) | 136 (50) | 64 (64) | 14 (41) | 0.020 |
| Diabetes mellitus (%) | 84 (31) | 55 (55) | 15 (44) | 0.0001 |
| COPD (%) | 14 (5) | 5 (5) | 1 (3) | 0.856 |
| ESRD (%) | 38 (14) | 10 (10) | 4 (12) | 0.594 |
| CHF (%) | 33 (12) | 25 (25) | 5 (15) | 0.009 |
| Prior history of CAD (%) | 28 (10) | 43 (43) | 8 (24) | 0.0001 |
| Prior stroke (%) | 15 (6) | 12 (12) | 6 (18) | 0.013 |
| cancer (%) | 68 (25) | 17 (17) | 5 (15) | 0.147 |
| Chronic medication use | | | | |
| ACEI/ARB | 43 (16) | 40 (40) | 6 (18) | 0.0001 |
| Beta-blocker | 38 (14) | 35 (35) | 4 (12) | 0.0001 |
| Statin | 18 (7) | 37 (37) | 1 (3) | 0.0001 |
| OHA | 35 (13) | 44 (44) | 10 (29) | 0.0001 |

Values are given as mean and standard deviation or numbers and percentages.

APACHE, acute physiology and chronic health evaluation; SOFA, sequential organ failure assessment; CRP, C reactive protein; COPD, chronic obstructive pulmonary disease; ESRD, end stage renal disease; CHF, congestive heart failure; CAD, coronary artery disease; ACEI, angiotensin converting enzyme inhibitor; ARB, angiotensin receptor blocker; OHA, oral hypoglycemic agent; ATT, antithrombotics; ICU, intensive care unit.

groups; neither did the occurrence of in-hospital gastrointestinal tract bleeding or AKI; in-hospital, 30-day; or ventilator and dialysis dependence. However, one-year mortality was found to be significantly lower in patients with chronic prescription of antithrombotic medication (Table 5). In cox regression model in patients with elevated troponin I, chronic antithrombotic treatment was associated with lower one-year all-cause mortality after adjusting age, gender, APACHE score, SOFA score, ICU admission diagnosis of shock or post operation status, focus of infection (pneumonia), initial serum lactate level, co-morbidities (tumor), co-administration of long term medication (beta blocker). (Table 6) The 30-day and one-year survival rates

**Table 5. Outcomes according to medications in patients with elevated troponin I.** dx.doi.org/10.17504/protocols.io.bfbmjik6.

| | No prescription (n = 249) | Chronic prescription (n = 100) | New prescription (n = 34) | P value |
|---|---|---|---|---|
| **Short-term outcomes** | | | | |
| Length of ICU stay | 12 + 7 | 11 + 7 | 10 + 5 | 0.276 |
| Length of hospitalization | 36 + 37 | 33 + 25 | 46 + 75 | 0.256 |
| GI bleeding during admission | 178 (65) | 67 (67) | 24 (71) | 0.802 |
| AKI during admission | 108 (40) | 45 (46) | 14 (41) | 0.594 |
| Hospital mortality (%) | 113 (41) | 26 (26) | 17 (50) | 0.009 |
| 30-days mortality (%) | 76 (28) | 19 (19) | 13 (38) | 0.063 |
| **Long-term outcomes** | | | | |
| Ventilator dependent (%) | 14 (5) | 6 (6) | 3 (9) | 0.669 |
| Dialysis dependent (%) | 34 (13) | 12 (12) | 2 (6) | 0.532 |
| One-year mortality (%) | 134 (49) | 34 (34) | 19 (56) | 0.017 |

Values are given as mean and standard deviation or numbers and percentages.

ICU, intensive care unit; GI, gastrointestinal; AKI, acute kidney injury.

in the troponin I elevation group, stratified by chronic and new prescription of antithrombotic are shown in Fig 3. These results showed a significant protective effect of chronic antithrombotic use for one-year survival ($p = 0.008$, log-rank test).

## Discussion

This study demonstrated that elevated troponin I at the time of ICU admission in the absence of ACS was associated with higher 30-day and one-year mortality. For patients with elevated troponin I levels, new antithrombotic prescriptions during ICU stay were not associated with 30-day or 1-year survival benefit whereas chronic antithrombotic use was related to better one-year survival. Although troponin I is a relevant biomarker for the diagnosis of AMI, troponin I elevation may be related to other conditions, such as heart failure, renal failure, left ventricular hypertrophy, and severe sepsis. [13] These conditions increased myocardial oxygen demand from elevated left ventricular end diastolic pressure, tachyarrhythmia or anemia. However, in patients with previous stable coronary artery disease, there was insufficient blood flow to meet increased myocardial oxygen demand which subsequently lead to myocardial injury. [14] Troponin I elevation was found in approximately 43% of patients admitted to non-cardiac ICUs in previous studies. [3] Troponin I elevation was more prevalent in our study, which can be explained by the difference in disease severity between our population and those reported on previously (mean APACHE II and SOFA scores were 27 and 9 in our study).

In-hospital and long-term prognostic impacts of troponin I elevation in these patients have been reported, and have been related in some, but not all, studies to poor prognosis. [3, 15, 16] We found that troponin I elevation was associated significantly with increased 30-day and one-year mortality. Moreover, Mohammed et al. [17] reported that troponin I elevation was associated with prolonged mechanical ventilation. Differences among studies are attributable to the inclusion of only patients with severe sepsis and septic shock in previous studies, whereas our study population comprised all patients admitted to the non-cardiac ICU with various medical and surgical conditions. Patients in elevated troponin I group had more complex disease status (i.e., higher APACHE II scores, advanced shock) and multiple co-morbidities (e.g., congestive heart failure, poorer renal function) relative to those without troponin I elevation. Our multivariate analysis showed that poorer prognosis was independent of the

**Table 6. Multivariate analysis of risk of one-year all-cause mortality of patients with elevated troponin I by a Cox regression model.** dx.doi.org/10.17504/protocols.io.bfbnjime.

| | Univariate analysis | | Multivariate analysis | |
|---|---|---|---|---|
| | HR (95% CI) | P value | HR (95% CI) | P value |
| Age (year) | 1.011 (1.002–1.020) | 0.020 | 1.008 (0.998–1.018) | 0.106 |
| Sex (male) | 1.315 (0.971–1.781) | 0.077 | 1.293 (0.923–1.812) | 0.136 |
| APACHE II | 1.057 (1.038–1.077) | 0.0001 | 1.051 (1.025–1.078) | 0.0001 |
| SOFA | 1.089 (1.042–1.138) | 0.0001 | 1.021 (0.963–1.083) | 0.491 |
| Admission diagnosis | | | | |
| Respiratory failure | 1.134 (0.850–1.512) | 0.392 | | |
| hemodynamic unstable | 1.433 (1.067–1.924) | 0.017 | 1.012 (0.723–1.418) | 0.944 |
| Post operation | 0.345 (0.196–0.606) | 0.0001 | 0.439 (0.231–0.835) | 0.012 |
| Focus of infection | | | | |
| Pneumonia | 1.575 (1.179–2.105) | 0.002 | 1.182 (0.843–1.657) | 0.331 |
| Bacteremia | 1.172 (0.769–1.787) | 0.459 | | |
| Intraabdominal infection | 1.000 (0.656–1.524) | 1.000 | | |
| Lab findings | | | | |
| Initial troponin I (ng/ml) | 1.018 (0.987–1.050) | 0.248 | | |
| Serum creatinine (mg/dl) | 0.938 (0.873–1.008) | 0.080 | | |
| HCO3 | 1.028 (0.998–1.059) | 0.065 | | |
| Lactate (mg/dl) | 1.005 (1.000–1.009) | 0.047 | 1.000 (0.994–1.005) | 0.970 |
| CRP (mg/dl) | 0.981 (0.959–1.004) | 0.105 | | |
| Underlying disease | | | | |
| Hypertension | 1.012 (0.759–1.348) | 0.937 | | |
| Diabetes mellitus | 0.904 (0.671–1.217) | 0.505 | | |
| ESRD | 0.927 (0.594–1.446) | 0.738 | | |
| CHF | 1.253 (0.864–1.817) | 0.234 | | |
| CAD | 0.866 (0.619–1.211) | 0.401 | | |
| CVD | 0.791 (0.459–1.363) | 0.398 | | |
| Cancer | 1.487 (1.077–2.052) | 0.016 | 1.528 (1.086–2.152) | 0.015 |
| ACEI/ARB, chronic | 0.983 (0.695–1.390) | 0.922 | | |
| Beta-blocker, chronic | 0.521 (0.339–0.800) | 0.003 | 0.625 (0.395–0.988) | 0.044 |
| Statin, chronic | 0.688 (0.432–1.094) | 0.114 | | |
| OHA, chronic | 0.843 (0.586–1.213) | 0.357 | | |
| Antithrombotic medication | | | | |
| Chronic vs. no | 0.584 (0.401–0.851) | 0.005 | 0.595 (0.391–0.906) | 0.015 |
| New vs. no | 1.198 (0.741–1.937) | 0.462 | 1.260 (0.742–2.140) | 0.392 |
| Chronic vs. new | 0.487 (0.278–0.855) | 0.012 | 0.472 (0.256–0.869) | 0.016 |

HR, hazard ratio; CI, confidence interval; APACHE, acute physiology and chronic health evaluation; SOFA, sequential organ failure assessment; CRP, C reactive protein; ESRD, end stage renal disease; CHF, congestive heart failure; CAD, coronary artery disease; CVD, cerebrovascular disease; ACEI, angiotensin converting enzyme inhibitor; ARB, angiotensin receptor blocker; OHA, oral hypoglycemic agent

APACHE II score, serum lactate level, underlying diseases (e.g., prior stroke, cancer), and prescription of beta-blockers and chronic antithrombotic agents.

Nathaniel et al. [18] reported that only 10% of patients with sepsis and elevated troponin I levels underwent invasive procedures, such as coronary angiography, of whom 39% received coronary revascularization with percutaneous coronary intervention or bypass surgery. That real-world, nationwide study reflects daily clinical practice. Most cardiologists commonly defer invasive management for patients with elevated troponin I levels with no clinical

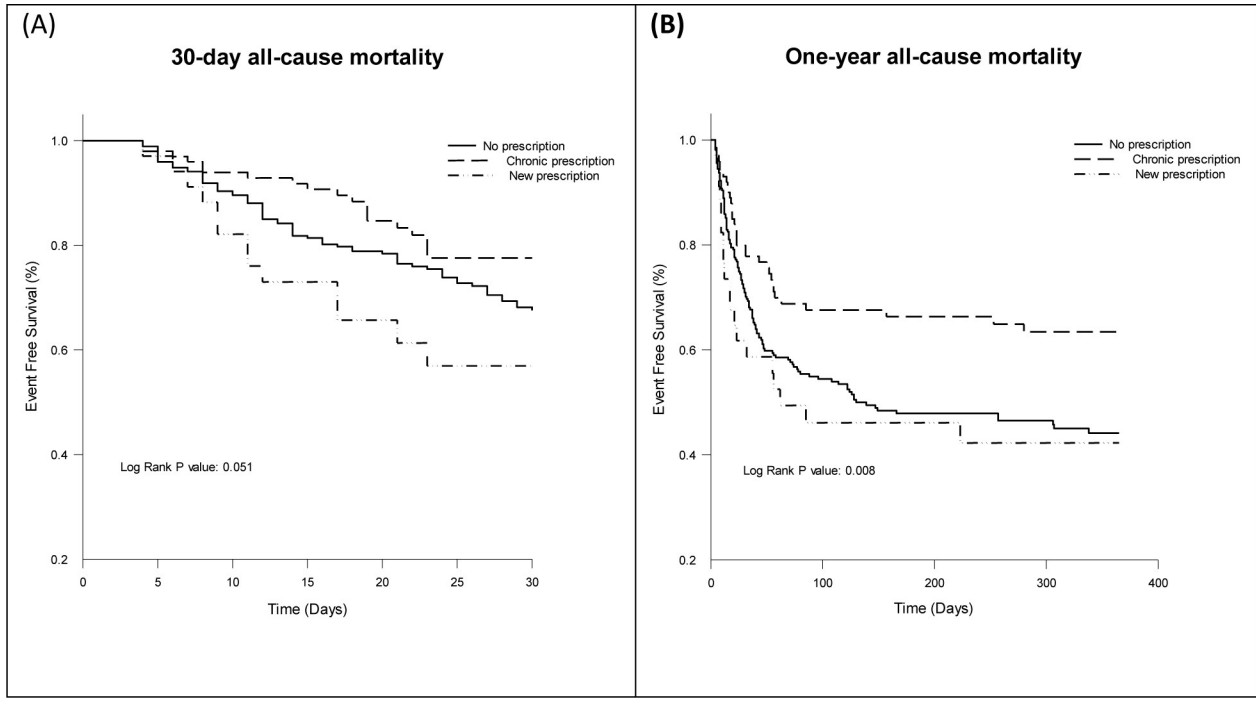

**Fig 3.** Kaplan-Meier curve for all-cause mortality according to antithrombotics use during ICU admission (A) 30-day all-cause mortality (B) one-year all-cause mortality. dx.doi.org/10.17504/protocols.io.bfbrjim6.

symptom, sign, or electrocardiographic evidence, primarily when type 2 MI is highly suspected. In our study, 8% of patients with elevated troponin I levels underwent coronary angiography and revascularization.

Medical therapy may be the mainstay of treatment for patients with type 2 MI. In the CURE study, combination therapy with aspirin and clopidogrel or ticagrelor resulted in well-established reductions in cardiovascular death and MI recurrence in patients with AMI, regardless of coronary revascularization receipt. [7, 19] Anti-coagulants such as low-molecular-weight heparin, enoxaparin, and unfractionated heparin further reduced the occurrence of adverse events. [20] However, the efficacy of these medications in patients with type 2 MI remains uninvestigated. Saraschandra et al. found that on-admission and delta troponin T levels were independent predictors of in-hospital and 1-year survival. Prior aspirin use in these patients did not change in-hospital or long-term mortality in analyses adjusted for age, sex, BMI, Charlson comorbidity index, APACHE III score, AKI, and respiratory failure. [21] The Platelet Glycoprotein IIb/IIIa in Unstable Angina Receptor Suppression Using Integrilin Therapy trial showed that patients with insignificant CAD are unlikely to benefit from glycoprotein IIb/IIIa therapy. [22] In ARRIVE trial, aspirin failed to reduce major cardiovascular events for primary prevention in patients with moderate cardiovascular risk as increased risk of bleeding. [23]

However, chronic antithrombotic medication reduced one-year all-cause mortality in patients with elevated troponin I levels. It may be partially explained by protective effect of these medication on the underlying diseases. Atrial fibrillation increased risk of stroke and sudden cardiac death in general population and those with history of coronary artery disease, myocardial infarction and heart failure. [24] In previous meta-analysis, warfarin reduced risk of stroke and thromboembolic complication from atrial fibrillation compared to non-warfarin

user and direct oral anticoagulation was superior to warfarin in prevention of stroke. [25] Subsequently, these anticoagulants lowered all-cause mortality in patients with atrial fibrillation possibly from preventing fatal ischemic stroke. [26, 27] Antiplatelet agents such as aspirin, Plavix, prasugrel and ticagrelor were found to have protective effect in patients with known occlusive vascular events such as myocardial infarction, ischemic stroke and peripheral arterial disease. These medications lowered risk of further major adverse cardiovascular events which may reduce all-cause mortality. [6, 28]

In a previous study, 4.7% of patients admitted to the ICU developed at least one episode of overt gastrointestinal bleeding, and 2.6% experienced clinically important gastrointestinal bleeding which subsequently increased risk of mortality. [29, 30] However, prevalence of overt gastrointestinal bleeding was not altered by treatment of aspirin or anticoagulant medication during ICU stay in previous cohort study (Odd ratio: 0.76, 95% CI: 0.35–1.64) and (Odd ratio: 2.05, 95% CI: 0.89–4.73) respectively as coagulopathy, acute kidney injury or co-existing liver disease were independent risk factors for clinically important GI bleeding in these patients apart from adverse effect of anticoagulant. [29] Nevertheless, the occurrence of gastrointestinal bleeding did not differ between patients treated with and without antithrombotics in our study, preventing examination of the potential for this factor to confound the primary outcome of all-cause mortality.

Our study has several limitations. It was retrospective and conducted at a single medical center. In addition, it was conducted in a non-cardiac ICU, where blood samples were not submitted for troponin I detection for all patients, which may have introduced selection bias. We included a diverse group of patients, in whom troponin I elevation may not have been related to obstructive CAD. In addition, a relatively small percentage of patients in our study underwent invasive coronary revascularization, which might have influenced the outcomes. Moreover, medication use in these critically ill patients may have been limited by coexisting medical conditions, as physicians may hesitate to prescribe antithrombotic agents, especially to patients with active gastrointestinal bleeding.

## Conclusion

Troponin I elevation at the time of non-cardiac ICU admission was associated with higher short-term and long-term mortality in critically ill patients. New prescription of antithrombotic medication to patients with elevated troponin I levels during their ICU stays showed no benefit, but chronic antithrombotic agent use was associated with a better one-year survival rate, suggesting that these drugs play a protective role in this high-risk population.

## Author Contributions

**Conceptualization:** Yi-Lin Tsai, Ming-Ren Kuo, Jiun-Yu Guo, Chi-Ting Lu, Chin-Sung Kuo, Po-Hsun Huang.

**Data curation:** Ya-Wen Lu, Yi-Lin Tsai, Ming-Ren Kuo, Po-Hsun Huang.

**Formal analysis:** Chi-Ting Lu, Po-Hsun Huang.

**Investigation:** Chuan-Tsai Tsai, Ya-Wen Lu, Ruey-Hsing Chou.

**Methodology:** Chuan-Tsai Tsai, Ya-Wen Lu, Ruey-Hsing Chou.

**Project administration:** Po-Hsun Huang.

**Supervision:** Ruey-Hsing Chou.

**Writing – original draft:** Chuan-Tsai Tsai, Ya-Wen Lu.

**Writing – review & editing:** Chuan-Tsai Tsai, Ya-Wen Lu.

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
