## [Decision Letter · Decision Letter 0]

31 Mar 2020

PONE-D-20-03834

Associations of antithrombotic agent use with clinical outcomes in critically ill patients with detectable troponin I in the absence of acute coronary syndrome

PLOS ONE

Dear Dr. Huang,

Thank you for submitting your manuscript to PLOS ONE. After careful consideration, we feel that it has merit but does not fully meet PLOS ONE’s publication criteria as it currently stands. Therefore, we invite you to submit a revised version of the manuscript that addresses the points raised during the review process.

We would appreciate receiving your revised manuscript by May 15 2020 11:59PM. To enhance the reproducibility of your results, we recommend that if applicable you deposit your laboratory protocols in protocols.io, where a protocol can be assigned its own identifier (DOI) such that it can be cited independently in the future. For instructions see: http://journals.plos.org/plosone/s/submission-guidelines#loc-laboratory-protocols

We look forward to receiving your revised manuscript.

Kind regards,

Corstiaan den Uil

Academic Editor

PLOS ONE

Journal Requirements:

2. Please include your tables as part of your main manuscript and remove the individual files. Please note that supplementary tables (should remain/ be uploaded) as separate "supporting information" files

4. Your ethics statement must appear in the Methods section of your manuscript. If your ethics statement is written in any section besides the Methods, please move it to the Methods section and delete it from any other section. Please also ensure that your ethics statement is included in your manuscript, as the ethics section of your online submission will not be published alongside your manuscript.

Reviewers' comments:

Reviewer's Responses to Questions

**Comments to the Author**

1. Is the manuscript technically sound, and do the data support the conclusions?

Reviewer #1: Partly

2. Has the statistical analysis been performed appropriately and rigorously? 

Reviewer #1: Yes

3. Have the authors made all data underlying the findings in their manuscript fully available?

Reviewer #1: Yes

4. Is the manuscript presented in an intelligible fashion and written in standard English?

Reviewer #1: Yes

5. Review Comments to the Author

Reviewer #1: Re: PONE-D-20-03834

Associations of antithrombotic agent use with clinical outcomes in critically ill patients with detectable troponin I in the absence of acute coronary syndrome

The authors report a cohort of ~600 mixed medical/surgical ICU patients without primary cardiac diagnoses, the majority of who had sepsis and respiratory failure. They examined the characteristics and outcomes of those with and without detectable cardiac troponin I and the associations between use of antithrombotic therapies (including antiplatelet and anticoagulant drugs) with outcomes. As in prior studies, patients with detectable troponin levels were sicker and had worse outcomes. Patients who received chronic antithrombotic therapy had lower mortality, although patients who received new antithrombotics did not.

Specific comments:

Introduction:

1. The authors discuss Type 1 and type 2 MI at the end of the opening paragraph without defining these entities and emphasizing their differences. Then, at the beginning of the secondary paragraph, it is implied that obstructive CAD is the difference between type 1 and type 2 MI, which is not true per se. The authors should be consistent about these distinctions, and introduce the distinction between type 1 and 2 MI earlier in the manuscript.

Methods:

1. The authors state that the upper reference limit of their troponin I assay is 0.16, so why did they choose >0.1 as their cut-off to define an elevated troponin. This should be justified, and the authors should use consistent terminology throughout the manuscript (detectable versus higher versus elevated). In addition, the brand/name of the assay should be provided.

2. The authors define sepsis as an increase in the SOFA score—I presume this also requires the presence of known or suspected infection, and this should be stated explicitly.

3. The authors do not state how they specifically excluded ACS patients, and this should be clarified.

4. Propensity adjustment or inverse probability weighting would be a better method to account for the baseline differences between antithrombotic treatment groups

Results:

1. The section “Antithrombotic use during ICU stays in patients with troponin I elevation” is confusing in places. At times, all antithrombotics are considered together (chronic and new), and at other times these are separated. This should be stated more clearly.

2. The statement “The lengths of ICU stay and hospitalization did not differ between the elevated and normal troponin I groups; neither did the occurrence of in-hospital gastrointestinal tract bleeding or AKI; in-hospital 30-day, or 1-year mortality; or ventilator and dialysis dependence (Table 5)” seems out of place here—by context, I presume the authors mean that the outcomes did not differ as a function of antithrombotic use but this should be clarified.

3. The analyses should be clearly performed and stated comparing acute and chronic antithrombotic use.

4. The section “Subgroup analysis of patients with detectable troponin I” likewise seems to be poorly named, as this examined the effects of several subgroups analyses and the results are not stated very clearly. If the authors’ main finding is that patients with elevated troponin who were chronically on anticoagulants had lower mortality, but patients who were started on new antithrombotics did not then this should be clearly stated and appropriate analyses performed to demonstrate the point (and Figure 2 revised to show new vs. old antithrombotics).

5. The authors do not clearly report whether any of their patients subsequently underwent echocardiography, stress testing, coronary angiography or PCI and these data should be provided.

Discussion:

1. Throughout, the authors should state that the different exposures (high troponin, receipt of antithrombotics) were ASSOCIATED with higher/lower mortality in this observational study, not using words or phrases that imply causation.

2. In the second paragraph, the authors discuss at length potential causes of type 2 MI and non-MI troponin elevations, and this discussion should be shortened as it is not directly relevant to their main research hypothesis.

3. The authors don’t really explain why chronic antithrombotic use is associated with better outcomes—they suggest that it might be related to the underlying disease without proposing a plausible mechanism of benefit.

4. The authors subsequently discuss GI bleeding at length even though this was unexpectedly not seen in their patients who received antithrombotics—this discussion should be shortened.

6. PLOS authors have the option to publish the peer review history of their article (what does this mean?). If published, this will include your full peer review and any attached files.

Reviewer #1: No

---

## [Author Response · Author response to Decision Letter 0]

29 Apr 2020

Responses to reviewer 1

Thank you very much for your valuable comments which are very instructive and helpful to this manuscript and our future research. The responses to the comments were dictated below and all changes to manuscript which mark with track changes.

1. Regarding the comment “The authors discuss Type 1 and type 2 MI at the end of the opening paragraph without defining these entities and emphasizing their differences. Then, at the beginning of the secondary paragraph, it is implied that obstructive CAD is the difference between type 1 and type 2 MI, which is not true per se. The authors should be consistent about these distinctions, and introduce the distinction between type 1 and 2 MI earlier in the manuscript.”

Ans: Thank you for your comment. The main difference between type 1 and type 2 MI is the pathophysiology in which type 1 MI was thrombus identification by angiography or intracoronary imaging or by autopsy, and the type 2 MI was contributed by supply and demand mismatch. As you suggested, we had added more descriptions in the introduction section. (Please see the page 5 / line 67.)

2. Regarding the comment “The authors state that the upper reference limit of their troponin I assay is 0.16, so why did they choose >0.1 as their cut-off to define an elevated troponin. This should be justified, and the authors should use consistent terminology throughout the manuscript (detectable versus higher versus elevated). In addition, the brand/name of the assay should be provided.”

Ans: Thank you for your comment. We agreed with you and corrected the mistake. We use the consistent terminology throughout the manuscript (all our data were based on troponin I > 0.1 as in all other places). (Please see the revised table 1 and 2)

As you suggested, we provided the brand/name of the assay in the manuscript. (Please see the page 7/ line 126.)

3. Regarding the comment “The authors define sepsis as an increase in the SOFA score - I presume this also requires the presence of known or suspected infection, and this should be stated explicitly.”

Ans: Thank you for your comment. We had stated the definition according to the Third international consensus definition for sepsis and septic shock. (JAMA 2016; 315 (8): 801 – 10). As you suggested, we mentioned this point in the revised Table 1.

4. Regarding the comment “The authors do not state how they specifically excluded ACS patients, and this should be clarified.”

Ans: Thank you for your comment. As you suggested, we excluded ACS patients in the current study and mentioned this point in the manuscript. (Please see the page 6/ line 85.)

5. Regarding the comment “Propensity adjustment or inverse probability weighting would be a better method to account for the baseline differences between antithrombotic treatment groups”.

Ans: Thank you for your comment. We agreed that propensity adjustment and inverse probability weighting are better methods in this situation. However, when considering the smaller number of patients in our study, we used the multivariate cox regression model to control baseline differences between the three groups. (Please see Table 6.)

6. Regarding the comment “The section “Antithrombotic use during ICU stays in patients with troponin I elevation” is confusing in places. At times, all antithrombotics are considered together (chronic and new), and at other times these are separated. This should be stated more clearly.”

Ans: Thank you for your comment. As you suggested, we divided the study patients with elevated troponin I to three groups according to the antithrombotic medications; patients with no prescription, chronic prescription, and new prescription during ICU stay. (Please see result section, page 8, line 133.)

7. Regarding the comment “The statement “The lengths of ICU stay and hospitalization did not differ between the elevated and normal troponin I groups; neither did the occurrence of in-hospital gastrointestinal tract bleeding or AKI; in-hospital 30-day, or 1-year mortality; or ventilator and dialysis dependence (Table 5)” seems out of place here - by context, I presume the authors mean that the outcomes did not differ as a function of antithrombotic use but this should be clarified.”

Ans: Thank you for your comment. As you suggested, the text is revised in Table 5. In previous cohort study (Intensive Care Med 2015; 41: 833 – 845), overt and clinical important GI bleeding were not increased by treatment with aspirin and anticoagulant. Relevant explanation was added in the discussion section, page 17, line 325.

8. Regarding the comment “The analyses should be clearly performed and stated comparing acute and chronic antithrombotic use.”.

Ans: Thank you for your comment. As you suggested, we divided the study patients with elevated troponin I to three groups according to the antithrombotic medications; patients with no prescription, chronic prescription, and new prescription during ICU stay. We also performed the Kaplan Meier Survival curve to compare the three groups. (Please see Figure 3.)

9. Regarding the comment “The section “Subgroup analysis of patients with detectable troponin I” likewise seems to be poorly named, as this examined the effects of several subgroups analyses and the results are not stated very clearly. If the authors’ main finding is that patients with elevated troponin who were chronically on anticoagulants had lower mortality, but patients who were started on new antithrombotics did not then this should be clearly stated and appropriate analyses performed to demonstrate the point (and Figure 2 revised to show new vs. old antithrombotics).”

Ans: Thank you for your comment. We agreed with you and revised the Figure 3 to clearly stated this point. (Please see the revised Figure 3.)

10. Regarding the comment “The authors do not clearly report whether any of their patients subsequently underwent echocardiography, stress testing, coronary angiography or PCI and these data should be provided.”

Ans: Thank you for your comment. As you suggested, we provided the information in Table 4. We agreed with you that coronary revascularization might alter the prognosis in these patients, which we mentioned in the limitation section. (Please see the page 15/ line 274.)

11. Regarding the comment “Throughout, the authors should state that the different exposures (high troponin, receipt of antithrombotics) were ASSOCIATED with higher/lower mortality in this observational study, not using words or phrases that imply causation.”

Ans: Thank you for your comment. As you suggested, we have rephrased our description to a softer tune in result section. (Please see page 12, line 216.)

12. Regarding the comment “In the second paragraph, the authors discuss at length potential causes of type 2 MI and non-MI troponin elevations, and this discussion should be shortened as it is not directly relevant to their main research hypothesis.”

Ans: Thank you for your comment. We agreed with you and have shortened the paragraph in discussion section. (Please see page 13, line 248.)

13. Regarding the comment “The authors don’t really explain why chronic antithrombotic use is associated with better outcomes - they suggest that it might be related to the underlying disease without proposing a plausible mechanism of benefit.”

Ans: Thank you for your comment. As you suggested, we have added more explanation in discussion section. (Please see page 16, line 306.)

14. Regarding the comment “The authors subsequently discuss GI bleeding at length even though this was unexpectedly not seen in their patients who received antithrombotics - this discussion should be shortened.”

Ans: Thank you for your comment. As you suggested, we have shortened the discussion section about GI bleeding accordingly. (Please see page 17, line 325.)

Thank you very much again! We appreciate your comments.

---

## [Editor Report · Decision Letter 1]

30 Apr 2020

Associations of antithrombotic agent use with clinical outcomes in critically ill patients with troponin I elevation in the absence of acute coronary syndrome

PONE-D-20-03834R1

Dear Dr. Huang,

We are pleased to inform you that your manuscript has been judged scientifically suitable for publication and will be formally accepted for publication once it complies with all outstanding technical requirements.

With kind regards,

Corstiaan den Uil

Academic Editor

PLOS ONE
---

## [Editor Report · Acceptance letter]

12 May 2020

PONE-D-20-03834R1 

Associations of antithrombotic agent use with clinical outcomes in critically ill patients with troponin I elevation in the absence of acute coronary syndrome 

Dear Dr. Huang:

I am pleased to inform you that your manuscript has been deemed suitable for publication in PLOS ONE. Congratulations! Your manuscript is now with our production department. 

With kind regards,

on behalf of

Dr. Corstiaan den Uil 

Academic Editor

PLOS ONE